# LightSail 2 Solar Sail Control and Orbit Evolution

**Justin R. Mansell** [1,*,†] ⬤, **John M. Bellardo** [2], **Bruce Betts** [3] ⬤, **Barbara Plante** [4] **and David A. Spencer** [1]

[1] Jet Propulsion Laboratory, California Institute of Technology, Pasadena, CA 91109, USA
[2] California Polytechnic State University, Department of Computer Science & Software Engineering, San Luis Obispo, CA 93407, USA
[3] The Planetary Society, Pasadena, CA 91101, USA
[4] Rogue Space Systems, Laconia, NH 03246, USA
[*] Correspondence: justin.r.mansell@jpl.nasa.gov
[†] Current address: 4800 Oak Grove Dr., Pasadena, CA 91109, USA

**Abstract:** The propellantless thrust of solar sails makes them capable of entirely new classes of missions compared to conventional or electric engines. Initiated in 2010, the Planetary Society's LightSail program sought to advance solar sail technology with the flights of LightSail 1 and 2. From launch in 2019 to deorbit in late 2022, LightSail 2 demonstrated the first controlled solar sailing in Earth's orbit using a CubeSat. By adjusting the orientation of the sail relative to the sun twice per orbit, LightSail 2 controlled solar radiation pressure on the sail to offset losses in orbital energy from atmospheric drag. Previous papers analyzed early mission results to show the effect this had on reducing the spacecraft's orbital decay rate. Subsequent refinements to the spacecraft's attitude control made throughout the mission eventually enabled it to achieve sustained net increases in orbital energy. This paper presents an analysis of the orbit changes and attitude control performance over the entire mission. Methods of assessing and improving the sail control are described. Activities and attitude behavior during the final deorbit phase are also analyzed, with results relevant to future drag sails as well as solar sail missions.

**Keywords:** LightSail; solar sailing; drag sail; attitude control; CubeSat; small satellite





## 1. Introduction

Solar sails stand apart from other forms of spacecraft propulsion that have been flown. By providing a large area with which to absorb and reflect photons from the Sun, they can achieve thrust without expending propellant. Although the thrust for typical sail designs is small, the continuous nature of this form of propulsion enables entirely new classes of trajectories. Missions have been proposed, for instance, based on a variety of non-Keplerian orbits, including pole-sitters and artificial Lagrange points [1,2]. Pole-sitters may be of considerable commercial and scientific interest for telecommunications and Earth remote-sensing, while artificial Lagrange points closer to the Sun than L1 could improve warning times for threatening space weather events [2,3].

Solar sails can also be an enhancing technology for conventional missions. They have featured prominently in work on solar–polar mission designs, as they can achieve the necessary inclination change without the Jovian gravity assist that drove the long-orbit period of the Ulysses mission [4–8]. Since the thrust of solar sails scales with the inverse square law, they are particularly advantageous for transfers and stationkeeping within the inner solar system. For example, they can be used to force sun-synchronous orbits at Mercury, where such orbits do not exist naturally yet are useful for remote sensing [9]. Trajectories that pass close to the Sun to harness the increased solar radiation pressure for achieving escape velocity have also been studied as a means of rapid transport to the outer solar system, heliopause, and interstellar space [2,10–12].

All of these potential missions rely on adjusting the orientation of the sail to control the magnitude and direction of the solar radiation pressure [13]. This can be especially

challenging for solar sails, since the sail is normally made of a large area of lightweight material to maximize the acceleration of the spacecraft. Demonstrating attitude control with a solar sail is therefore a key step in advancing the technology readiness level (TRL) of the technology.

The first step in this respect was taken by JAXA's IKAROS spacecraft [14]. The experiment used an array of reflectivity control devices to reverse the drift in the spacecraft's spin axis by 0.5 degrees in 24 h while en route to Venus in 2010. The acceleration provided by the sail was also measured via ground-based Doppler and showed a delta-V of approximately 100 m/s delivered during the first six months.

IKAROS had a wet mass of 310 kg and a sail area of 200 m$^2$ [14]. This resulted in a sail loading of 1467 g·m$^{-2}$. The next solar sail to fly was NASA's NanoSail D2 in 2011 [15]. Though uncontrolled and deployed on an Earth orbit too low to observe the effects of solar radiation pressure due to atmospheric drag, the combination of a 10 m$^2$ sail deployed from a 4 kg 3U CubeSat bus gave the spacecraft a sail loading of 400 g·m$^{-2}$. Sail loading is inversely related to metrics such as lightness number or characteristic acceleration, which broadly define the trajectory capabilities of the sail [1].

The compact nature of CubeSats therefore brings both advantages in sail performance and challenges in miniaturizing the attitude determination and control system (ADCS) to control the sail. Advancing the TRL of CubeSat solar sails was one of the main goals of The Planetary Society's LightSail program, which began crowdfunding the development of a pair of Earth-orbiting 3U solar sails in 2010. These became LightSail 1 and LightSail 2 (LS2).

LigthSail 1 flew in 2015 and demonstrated successful deployment and imaging of its 32 m$^2$ sail [16]. Like NanoSail D2, it was uncontrolled and launched into a low orbit (356 × 705 km) that decayed quickly after deployment of the sail. The goal of LightSail 2 was to demonstrate controlled solar sailing using a CubeSat. This has two aspects: demonstrating successful attitude control of the sail, and demonstrating that the attitude control has the expected effect on the orbit due to solar radiation pressure.

LS2 was launched into a 709 × 726 km altitude 24 degree inclination orbit on 25 June 2019. Its sail was deployed on 23 July 2019, and the spacecraft operated continuously until its eventual deorbit on 16–17 November 2022. The mission succeeded in demonstrating controlled solar sailing using a CubeSat. By reorienting the sail by 90 degrees twice each orbit, LS2 harnessed solar radiation pressure to slow, and sometimes even reverse, its orbit decay due to atmospheric drag.

Previous publications about the mission have focused on the design and early operations of the spacecraft [17–19]. However, continuing refinements to the spacecraft's ADCS improved attitude control performance and culminated in a period of sustained orbit energy increases during the summer of 2021. The purpose of this paper is to discuss these improvements and assess the ADCS and solar sailing performance over the full mission. Section 2 reviews the design and solar sailing concept of operations for LS2. Section 3 analyzes ADCS performance. Section 4 discusses the orbit evolution over the mission and the impact of sail control. Finally, Section 5 assesses the final week of operations to determine if the sail may have collapsed prior to re-entry. The results are relevant not only to future solar sails, but also to deployable deorbit devices that are becoming an increasingly popular means of space debris mitigation.

## 2. Attitude Determination and Control System Overview

### 2.1. Hardware

The main difference between LS2 and previous CubeSat solar sails was the inclusion of a 3-axis stabilized ADCS. This also made LS2 the first 3-axis stabilized sail to fly instead of the spin-stabilized type like IKAROS. Attitude knowledge was derived from two magnetometers, five Sun sensors, and one of two sets of gyros processed by an extended Kalman filter. The configuration of these sensors is shown in Figure 1.

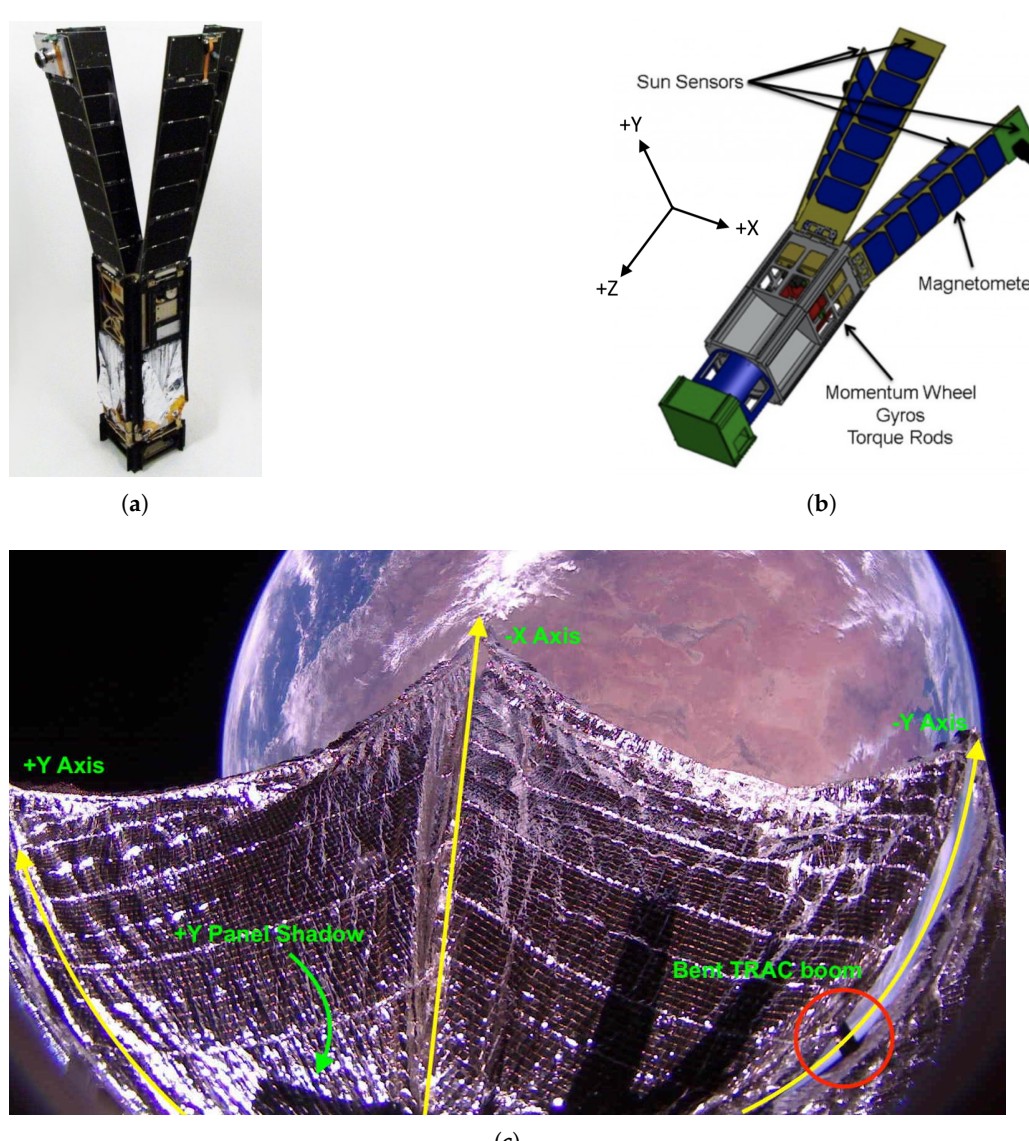

**Figure 1.** Configuration of the LightSail 2 spacecraft. (**a**) The flight unit with solar panels fully deployed. (**b**) ADCS axis definitions and placement of sensors. (**c**) Image of the solar sail on 15 January 2020 showing the shadows of the deployed panels and a bent support boom.

The magnetometers were fixed on the +X and +Y solar panels. A Sun sensor was located on the end of each deployable panel such that their 75° (half-cone) fields of view overlapped about the −Z axis. The fifth Sun sensor was located on a solar panel on the −Z face of the spacecraft body. A voting scheme was employed amongst the Sun sensors to reject measurements that were beyond the $3\sigma$ noise characteristic of the sensors. This was critical because the sensors were not configured to read the intensity of the source they were measuring. Thus, when one or more of the sensors could not see the Sun, voting prevented some or all of the sensors from being ingested by the attitude filter. Previous work early in the mission analyzed the effectiveness of this voting scheme [17].

Though not used for attitude determination, there were also two wide-angle cameras located on the +X and −X solar panels. These allowed the condition of the sail to be monitored and facilitated public outreach with images of Earth. The solar panels were initially stowed against the spacecraft but deployed at 155 degrees to achieve the configuration pictured in Figure 1a,b. An exception was the +Y panel. Starting in 2020, images such as Figure 1c revealed that the +Y panel had deployed only partially. Minimizing



the average angular discrepancy between the +X and +Y magnetometer measurements in the spacecraft body frame required a rotation of 92° from the stowed configuration. This correction was subsequently incorporated into the sensor-to-body frame rotation matrices for +Y magnetometer and sun sensor. To compute residuals for the attitude Kalman filter, measurements were interpreted according to either a solar position model or the 2010 (later 2020) International Geomagnetic Reference Field (IGRF). The spacecraft position with which to query the IGRF for the local magnetic field vector was provided by two-line elements (TLEs) that were uplinked to LS2 several times per week and propagated onboard to the time of the spacecraft clock using the Simplified Generalized Perturbations (SGP4) dynamic model.

To supplement the Sun sensors and magnetometers, angular rates were measured by either the mainboard gyros or a set of three primary (PIB) gyros (one for each body axis). The mainboard gyros were uncalibrated and only used in spacecraft modes that did not require attitude knowledge. The PIB gyros required higher power but provided rates to an accuracy of $\sigma = 0.27$ deg/s.

Attitude control was accomplished with a magnetic torque rod for each axis and a momentum wheel actuating about the +Y axis. The wheel enabled turns about the Y axis to be conducted on the order of a few minutes, while the torque rods stabilized the other axes and performed momentum management. Finally, the sail itself was composed of 4 triangular sheets of Mylar extended by 4 m triangular rollable and collapsible (TRAC) booms upon deployment, for a total area of 32 m$^2$. The Mylar had an aluminum coating on the $-Z$ side of the sail.

Tables 1 and 2 summarize the essential ADCS components and characteristics of the spacecraft. The ADCS control loop operated at 1 Hz. Telemetry was logged once per 5 min, but could briefly be collected at up to 0.2 Hz upon command.

**Table 1.** LightSail 2 ADCS components.

| Component | Manufacturer | Metric | Value |
|---|---|---|---|
| Magnetometers | Honeywell | $1\sigma$ noise | 0.2 µT |
| Sun sensors | Elmos | Resolution | 2.7 deg |
| Primary gyros | Analog Devices | $1\sigma$ noise | 0.27 deg/s |
| Momentum wheel | Sinclair Interplanetary | Max RPM | 5920 |
| | | Max Torque | $5\times 10^{-3}$ N·m |
| | | Max Angular Momentum | 0.06 N·m·s |
| Torque rods | Stras Space | Max Dipole | 1 A·m$^2$ |
| | | Duty Cycle | 700 ms/s |

**Table 2.** LightSail 2 spacecraft characteristics.

| Design Detail | Value |
|---|---|
| Mass | 4.93 kg |
| Sail area | 32 m$^2$ (square) |
| Sail loading | 154 g·m$^{-2}$ |
| Volume of bus | 3U ($10 \times 10 \times 30$ cm) |
| Moments of inertia | $I_x = 3.79$ kg·m$^2$, $I_y = 3.79$ kg·m$^2$, $I_z = 7.33$ kg·m$^2$ |
| Products of inertia | $I_{xy} = -1.90\times10^{-4}$ kg·m$^2$, $I_{xz} = -8.18\times10^{-4}$ kg·m$^2$, $I_{yz} = 1.47\times10^{-3}$ kg·m$^2$ |

## 2.2. Control Strategy

The ADCS software included 6 operational modes that are described in Table 3. All modes used an extended Kalman filter for attitude determination. Depending on the mode, either the mainboard or PIB gyros were processed by the filter alongside the magnetometers and Sun sensors. Separate from the Sun sensor voting scheme, the filter was programmed to ignore the Sun sensors whenever the onboard TLE propagation indicated that LS2 was

in eclipse. Detumble mode (mode 0) used a B-dot algorithm to command the torque rods to oppose the spacecraft rotation, detected by differencing successive magnetometer measurements in the spacecraft body frame. It could therefore operate even with poor attitude knowledge from the Kalman filter. Mode 1 was similar but sacrificed rate damping about the Z axis to provide alignment with the local magnetic field. Modes 2, 4, and 5 were actively controlled modes. They generated a command quaternion based on the spacecraft velocity and inertial position of the Sun. A PD controller then generated command torques from each actuator in order to align with desired quaternion. An accurate quaternion estimate from the Kalman filter was therefore essential to these modes.

The most frequently used mode and the most important for the overall mission was solar sailing mode (mode 2). In this mode, LS2 slewed between two attitudes, as depicted in Figure 2. When moving away from the Sun, LS2 pointed the sail normal vector (+Z) anti-parallel to the Sun. This was the "On" attitude and served to maximize the solar radiation pressure on the sail and project the thrust in a direction that increased the orbit energy. The "Off" attitude on the other half of the orbit turned the sail to minimize thrust so that this energy gain was not undone when moving towards the Sun. Thus, the On–Off strategy allowed solar radiation pressure on the sail to contribute an increase in the orbit energy that could oppose orbit decay due to atmospheric drag. While not the mathematically optimal maneuvering strategy for raising orbit energy, the On–Off strategy simplified ADCS by holding a fixed attitude between slews [13].

The other frequently used modes were detumble (mode 0) and no-torques (mode 3). For most of the mission, LS2 transitioned into detumble mode once per day for two orbits followed by one orbit in no-torques. The primary purpose of this was momentum management. Transitioning to mode 0 engaged regenerative braking on the reaction wheel, transferring its stored angular momentum into the spacecraft rotation rates. Detumble mode could then dampen these rates using the torque rods, removing momentum from the spacecraft. Since the attitude was necessarily allowed to tumble in this mode, solar power was less consistent than in mode 2. The short period in no-torques mode before transitioning back to solar sailing allowed the spacecraft batteries to recharge before re-engaging the momentum wheel. This was based on the experience that spinning up the momentum wheel could drain power and cause a spacecraft reboot if mode 2 was engaged directly from detumble mode. Several other momentum management strategies were trialed early in the mission and are described in [17], but this strategy proved the most successful.

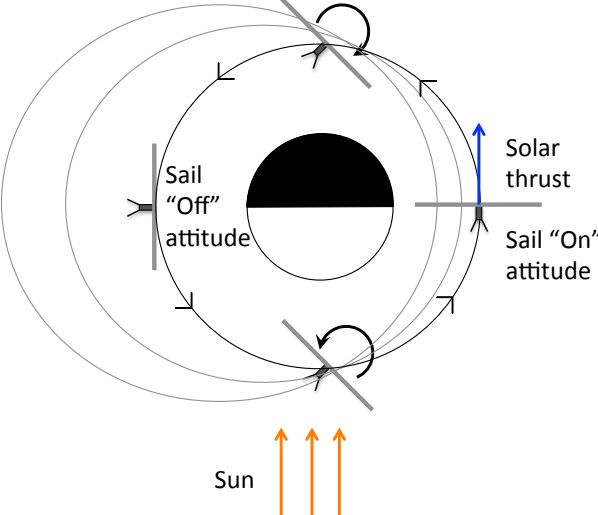

**Figure 2.** LS2 orbit raising strategy.

The remaining spacecraft modes were rarely used. Mode 1 was used to provide good attitude for communication during spacecraft commissioning. Mode 4 was a pure "On"

attitude that always pointed +Z away from the Sun, and mode 5 was a velocity pointing mode that was never tested successfully due to the constantly slewing attitude it required.

**Table 3.** LightSail 2 attitude control modes.

| Mode | Purpose | Torque Rods | Momentum Wheel | PIB Gyros |
|---|---|---|---|---|
| 0 | Detumble: Magnetometers and B-dot control used to generate torque rod commands to oppose the spacecraft's rotation. | ✓ | - | - |
| 1 | Magnetic alignment: +Z axis torque rod set to constant maximum power while the others acted as in the detumble mode. This approximately aligned the +Z axis with the local magnetic field vector. | ✓ | - | - |
| 2 | Solar sailing: Two 90-degree slews commanded each orbit to follow the "On–Off" control strategy. | ✓ | ✓ | ✓ |
| 3 | No torques: All actuators powered off. | - | - | - |
| 4 | Sun pointing: +Z axis maintained anti-parallel to the Sun. | ✓ | ✓ | ✓ |
| 5 | Velocity pointing: +Z axis maintained parallel to inertial velocity. | ✓ | ✓ | ✓ |

## 3. Attitude Performance

### 3.1. Methods of Assessment

LS2's onboard attitude filter produced a quaternion estimate that was available in regularly downlinked telemetry. This quaternion represents a body frame to J2000 Earth Mean Equator rotation and is the best estimate for the spacecraft's attitude at each telemetry point. Also downlinked in the telemetry were the raw measurement values from each sensor and the latitude, longitude, and altitude of the spacecraft as determined by the onboard TLE propagator. The latter allows the inertial velocity of the spacecraft relative to the Sun to be reconstructed to determine at each telemetry point whether the "On" or "Off" sail orientation was being commanded. Note that it is important to derive these commands from positions calculated by the onboard TLE propagator, since this is what the spacecraft used when determining whether to command either the "On" or "Off" attitude. Spacecraft clock errors or stale TLEs could sometimes lead to discrepancies in spacecraft position compared to TLEs propagated on the ground.

Once the sail command is reconstructed, it can be compared to the onboard quaternion estimate to determine the pointing accuracy of the spacecraft. The results were typically assessed using plots such as those shown in Figure 3. The plots show the actual orientation of the spacecraft compared against the On–Off commands at five select points in the mission. The first plot Figure 3a shows the slew performance during spacecraft commissioning before the solar sail was deployed. The next Figure 3b exemplifies how recognizable On–Off control was initially intermittent but improved as the momentum management strategy was refined. By mid-2020 (Figure 3c), an accumulating bias in the PIB gyros disrupted control. An in-flight recalibration strategy was developed in 2021 and restored sail control, ultimately leading to improved solar sailing and consistent increases in the semi-major axis (Figure 3d,e).

Since the pointing accuracy assessment is based on the spacecraft's own attitude estimate, it is important to assess the quality of that estimate. In theory, the Kalman filter provides a state covariance matrix alongside the estimated quaternion that is fit for this purpose, but it is based on assumptions of wide-sense stationary stochastics with mean zero white noise for measurement errors. In practice, the actual noise of the magnetometers was difficult to determine and likely varied with time due to interference from currents in the solar panels. Both the mainboard and PIB gyros were also corrupted by a time-varying bias that became increasingly problematic over the mission.

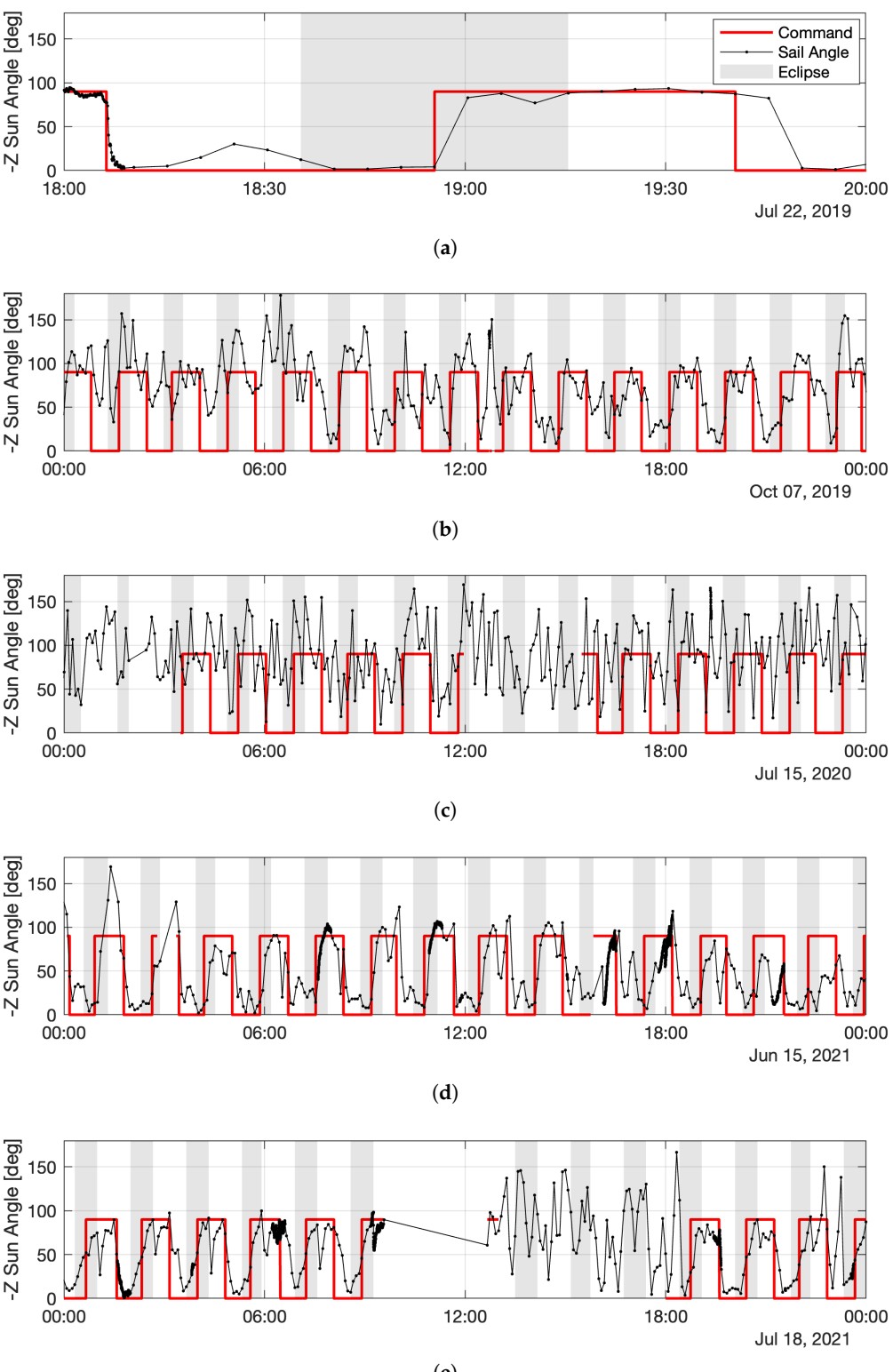

**Figure 3.** Selected dates of solar sailing. (**a**) Before sail deployment. (**b**) Solar sailing several months into the mission. (**c**) By mid 2020, successful sail control was no longer discernible due to an accumulating bias in the primary gyros. (**d**) In-flight recalibration of the gyros in 2021 restored performance and led to some of the best solar sailing of the mission. (**e**) Solar sailing on the day with the highest observed gain in orbit semi-major axis. Portions of the plots where the sail command is not visible correspond to momentum desaturation events during which the sail angle is uncontrolled.

We therefore developed an alternative "quaternion consistency check" with which to monitor the reliability of the onboard attitude. The check verifies that the estimated quaternion can reproduce the observed magnetic field by using the quaternion to rotate the IGRF magnetic field vector queried at the spacecraft's position into the spacecraft body frame. The angular error between the transformed IGRF vector and the average of both magnetometer measurements provides a crude but independent estimate of the attitude estimation error. While an angular error of zero does not guarantee robust attitude knowledge (errors in rotation about the magnetic field vector are not detectable), the check was nonetheless extremely useful in detecting periods of degraded attitude knowledge due to the IGRF being queried at an incorrect position.

Figure 4 depicts a schematic of the consistency check alongside plots showing the discrepancy angle during the initial recalibration of the PIB gyros and later during an anomaly. The anomaly resulted from the flight computer running out of memory, which halted the onboard orbit propagator and caused a loss of attitude knowledge.

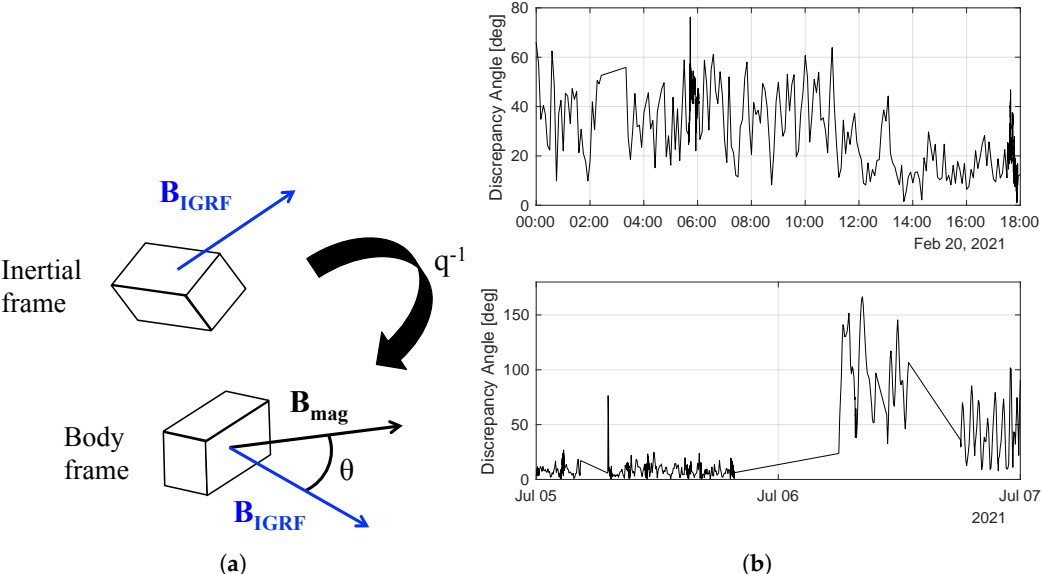

**Figure 4.** Quaternion consistency check. (**a**) The test rotates the IGRF magnetic field vector into the body frame using the onboard attitude estimate and calculates the discrepancy with the measured magnetic field vector. (**b**) (**Top**): the discrepancy angle immediately after the PIB gyros were recalibrated for the first time. (**Bottom**): discrepancy angle during a loss of attitude knowledge.

One final method by which ADCS performance was assessed was by detecting and diagnosing anomalies. Initially, this was done by manually inspecting telemetry or by aggregating data into derived signals such as the quaternion consistency check. Examples of some common anomalies were stuck values in one of the magnetometers, spacecraft clock resets, cyclic computer resets due to corrupted telemetry files, and shadowing of the solar panels by the sail. Beginning in 2020, we developed a machine learning model that was pre-trained on LS2's ADCS simulator and deployed as a ground tool to both detect and diagnose anomalies in LS2 telemetry. The system proved useful in identifying anomalies that may have otherwise gone unnoticed [20].

### 3.2. Performance over the Mission

Figures 3 and 4 provide a look at attitude knowledge and control performance at a few select points in the mission. To discuss trends over the entire mission, Figures 5 and 6 display distributions for the quaternion discrepancy and pointing error month by month. We discuss the history of LS2's attitude knowledge first.

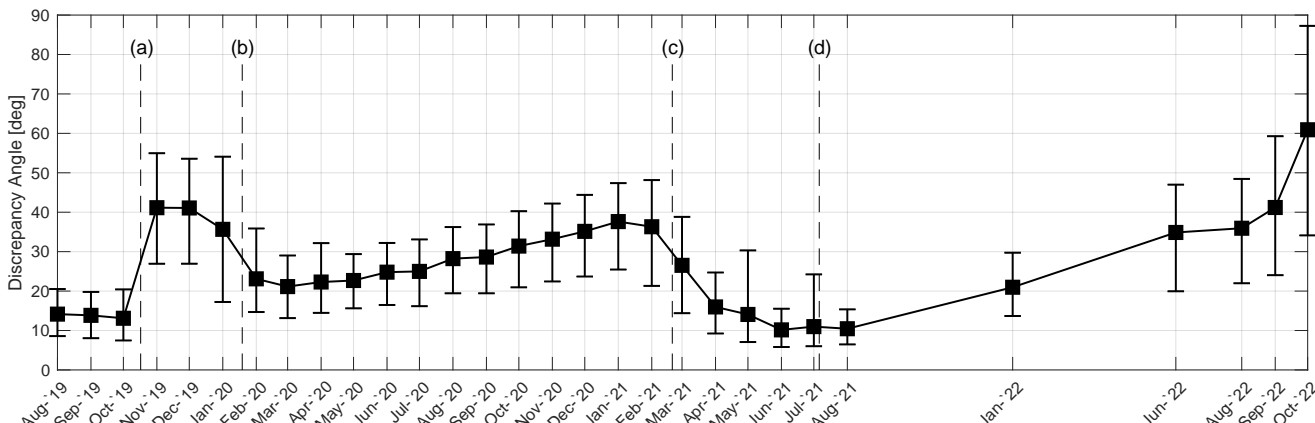

**Figure 5.** LightSail 2 monthly attitude knowledge measured by the 20th–80th percentiles of the quaternion consistency check. (a) First occurrence of the memory overload anomaly. (b) Incorporation of +Y sensors after measuring the +Y solar panel deployment angle. (c) Beginning of the gyro recalibration campaign. (d) Other occurrence of memory anomaly.

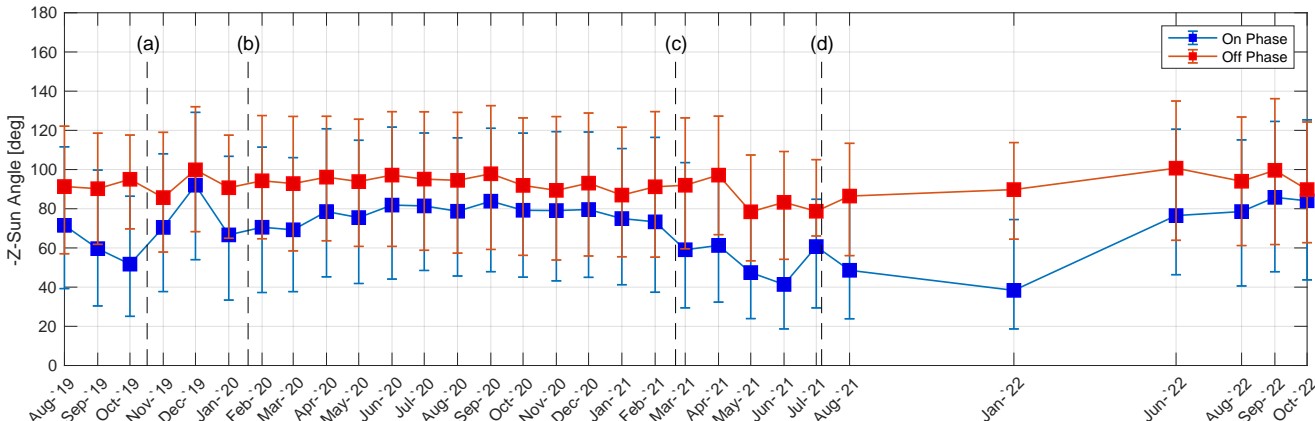

**Figure 6.** LightSail 2 monthly On–Off control performance represented by the 20th–80th percentiles of the −Z to Sun angles for the On and Off phases of each orbit. (a) First occurrence of the memory overload anomaly. (b) Incorporation of +Y sensors after measuring the +Y solar panel deployment angle. (c) Beginning of the gyro recalibration campaign. (d) Other occurrence of memory anomaly.

### 3.2.1. Attitude Knowledge

Figure 5 calculates the quaternion discrepancy for each 5-min telemetry sample from ADCS mode 2 in a given month and plots the median value along with error bars extending to the 20th and 80th percentiles. Each point therefore represents the distribution in attitude knowledge error for that month. Using the 20th and 80th percentiles helps filter out transient spikes, such as those following momentum desaturation events while the attitude filter re-converges. In some months not enough telemetry was downlinked to report reliable statistics. This was particularly true towards the end of the mission when the reduced availability of spare parts for tracking equipment due to the COVID-19 pandemic led to less consistent communication with the spacecraft. Months for which telemetry covers less than 15% of the month (about 4.5 days) are thus omitted from the plot.

Beginning on the left-hand side in August 2019, Figure 5 shows how attitude knowledge error was consistent at about 15 deg for the first several months of the mission. During these months, it was discovered that both the +Y sun sensor and magnetometer were behaving anomalously [17]. The anomalous deployment of the +Y solar panel was unknown at this time, so the sensors were simply passivated so as not to corrupt the attitude estimate. On 17 October 2019, at annotation (a) in the plot, LS2 encountered the overloaded memory anomaly for the first time. This anomaly occurred several times during the mission, often

because interruptions to regular communication delayed uplinking the commands to free memory. When this anomaly manifested, it caused a complete loss of attitude knowledge due to crashing the onboard TLE propagator. Resolving the anomaly required re-uplinking several critical files, one of which included the passivation instruction for the +Y sensors. The first time this anomaly was encountered, an earlier version of that file that did not include the passivation was uplinked, causing the +Y sensors to be ingested into the attitude filter with incorrect sensor-frame-to-body-frame rotation matrices. This was the cause of the degraded attitude performance throughout November and December 2019.

In January 2020, images such as Figure 1c revealed the anomalous deployment angle of the +Y solar panel. Further analysis refined the deployment angle, and the rotation matrices for the +Y sensors were updated, allowing them to be correctly incorporated into the attitude estimate following annotation (b). This is reflected in the improved attitude knowledge for January and February 2020.

Over the next year, attitude knowledge slowly degraded once again. By February 2021, the cause of this was isolated to an accumulating bias in the PIB gyros, and a recalibration strategy was implemented (see Section 3.3). This was an iterative process. Beginning at point (c) in the plot, a series of recalibrations restored the level of quaternion discrepancy to slightly better than the beginning of the mission. At point (d), the memory overload anomaly occured once again but was quickly resolved.

Over the following months until the end of the mission, tracking station difficulties greatly reduced the amount of telemetry being downlinked. This made further recalibrations of the PIB gyros difficult. As a result, the attitude knowledge degraded at a rate similar to 2020.

### 3.2.2. Sailing Performance

Figure 6 shows the month-to-month solar sail control. Similar to Figure 5, the plot collects the 20th–80th percentile range for the attitude pointing in each month. The angle between the spacecraft $-Z$ axis and the Sun is plotted separately for the On and Off phases of the control strategy. Recall that the On phase commands a $-Z$-Sun angle of 0 deg, while the Off phase commands 90 deg. The distance between the distributions for the On and Off angles in each month provides a measure of how well LS2 is following this strategy.

The control performance generally mirrors the trend in attitude knowledge seen in Figure 5. Performance improved initially as control gains and the momentum management strategy were refined, but degraded between points (a) and (b) due to poor attitude knowledge. Recognizable On–Off slews were intermittent in 2020 before disappearing altogether due to accumulating biases on the gyros. The recalibration of the gyros and the resulting improvement in attitude knowledge in 2021 at point (c) restored successful On–Off control. Additional recalibrations improved performance further throughout the spring and summer of 2021 and provided the best sailing of the mission. An overloaded memory briefly interrupted this period in early July but was resolved within several days. This is the cause of the brief spike in On-phase pointing error near point (d). Excellent control performance followed and persisted until at least January 2022. However, the subsequent slow degradation of attitude knowledge meant that this was one of the last periods of recognizable On–Off slewing. By the last two months before deorbit, LS2 was essentially tumbling freely.

A notable feature of the plot is that the median Off-phase pointing remains close to 90 deg even when attitude knowledge is poor, while the On-phase performance varies significantly throughout the mission. This is a reflection of the different pointing objectives. For the On-phase, only a single attitude points the -Z spacecraft axis towards the Sun, whereas for the Off-phase, an angle of 90 deg can be satisfied by any rotation about the Sun vector $\hat{s}$ that provides $-\hat{z} \cdot \hat{s} = 0$. A better way to assess the Off-phase performance is to consider the narrowness of the distribution about a 90 deg pointing angle. In months such as January 2022, the 20th–80th percentile range is much smaller—about 90 deg—than it is for months with poor sailing, such as September 2020.

### 3.3. On-Orbit Gyro Recalibration

Key to LS2's successful solar sailing during its extended mission was the recalibration of the primary gyros. By early 2021, both attitude knowledge and control performance were poor, leaving the satellite in an essentially uncontrollable attitude similar to Figure 3c. A simple visual inspection of PIB gyro telemetry by this time showed that the X-rotation rates were noticeably biased towards positive rates. This, along with data from the diagnosis tool, led us to hypothesize that the X axis and possibly other PIB gyros had accumulated significant bias.

In February 2021, we began a campaign of estimating these biases by cross-calibrating PIB rates against rates derived from the magnetometers. During periods of 5 s telemetry, the spacecraft rotation rate is essentially constant from one time step to the next. The angular rate can therefore be estimated from successive magnetometer measurements in the spacecraft body frame:

$$\theta_k = \sin^{-1}\left(\frac{\|\mathbf{b}_k \times \mathbf{b}_{k-1}\|}{\|\mathbf{b}_k\|\|\mathbf{b}_{k-1}\|}\right) \tag{1}$$

$$\hat{\boldsymbol{\omega}}_k = \frac{\mathbf{b_k} \times \mathbf{b_{k-1}}}{\|\mathbf{b_k} \times \mathbf{b_{k-1}}\|} \tag{2}$$

$$\boldsymbol{\omega}_k = \frac{\theta_k}{t_k - t_{k-1}}\hat{\boldsymbol{\omega}}_k. \tag{3}$$

Equation (1) calculates the angle traversed by the measured magnetic field vector **b** in the body frame between time steps k and k−1. Equation (2) computes the axis of that rotation. Equation (3) combines these into the angular rate vector of the body frame with respect to the inertial frame.

Although rates derived in this manner are too noisy to use for attitude control directly, over enough data points, they can provide a meaningful comparison against the PIB measurements. We also averaged rates derived independently from the +X and +Y magnetometers to compare against the rate determined from the PIB gyros at each time step.

Figure 7 shows the cross-calibration of all 5 s PIB telemetry with the corresponding components of the derived rates between 10 December 2020 and 5 February 2021. A linear fit confirmed a nearly 0.5 deg/s bias in the PIB X measurements while also revealing a 0.14 deg/s and 0.04 deg/s bias in the Y and Z gyros, respectively. The slopes of each line were neglected other than to verify that they were positive. We subsequently uplinked a software update to subtract these offsets from the PIB gyros. As shown in Figures 4b and 5c, the result was an immediate improvement in attitude knowledge. We continued to iterate on these estimates as additional 5 s telemetry was downlinked, uploading further corrections approximately once per month. The final bias update was performed in October 2022. The reduced availability of 5 s telemetry afterwards prevented further updates.

### 3.4. Momentum Management Strategy

Significant improvements in attitude control performance during the first three months of solar sailing were achieved through refinements to the angular momentum management strategy. At deployment, momentum desaturation ("desats") was commanded manually via transition to mode 0 followed by a manual command back to solar sailing. This sometimes led to long periods of momentum wheel saturation if the saturation occurred shortly after the final tracking pass for a day, or if mode 2 could not be re-commanded in time before end of the last pass.

The next strategy was to automatically command 10 min in mode 0 if the wheel speed exceeded 4000 RPM. This was performed by an onboard script. The strategy was retired after less than a month, however, as it was observed that the frequent transitions were disruptive to attitude control, as LS2 usually required multiple orbits to reacquire a controlled attitude after tumbling in mode 0. The final strategy that was employed was to adjust the onboard script to automatically command mode 0 for two orbits at a regular

cadence regardless of the wheel RPM. This strategy also introduced the delay in mode 3 before cycling back to solar sailing. Initially, the desats were performed every day with only 5 min in mode 3. By 2021, the desats were being commanded once every two days with a full orbit in mode 3. This remained the strategy for the rest of the mission.

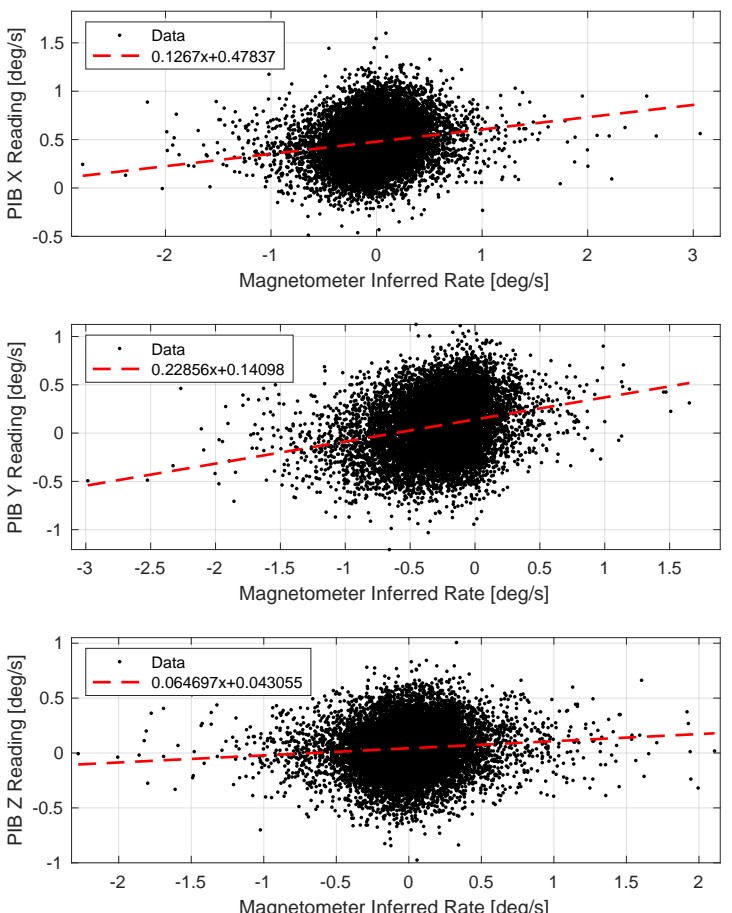

**Figure 7.** Cross-calibration of LightSail 2's primary gyros against rates derived from the magnetometers. The y-intercept of each line of best fit provides an estimate of the bias accumulated by each gyro.

## 4. Orbit Evolution

### 4.1. Overview

Having discussed both attitude determination and control, we now turn our attention to how LS2's orbit changed throughout the mission. The sources for orbit knowledge both during and after the mission are TLEs produced by the 18th Space Control Squadron and distributed via Celestrak. These were typically published about once or twice per day and allowed for a calculation of LS2's state vector and Brouwer–Lyddane mean orbital elements at epoch. Figure 8 plots the perigee, apogee, and altitude of the semi-major axis (reference Earth radius: 6378 km) derived from the first LS2 TLE to the last.

Two features are notable in Figure 8a. The first is the orbit decay from the initial mean altitude of 717 km. Beginning at sail deployment, the altitude trends downwards until deorbit. Although the decay rate generally accelerated as LS2 dropped deeper into the atmosphere, it was variable and not monotonic. Large jumps in the decay rate were often observed during solar storms such as those caused by Coronal Mass Ejections. The decay rate also varied in response to solar sailing performance. This effect was most visible between June and August 2021, where the semi-major axis rose by about 750 m and coincided with the best solar sailing of the mission. Figure 8b shows a zoomed-in view of

the semi-major axis during this period. Section 4.2 analyzes the orbit raising in more detail, while Section 4.3 looks at the impact of solar activity on the orbit.

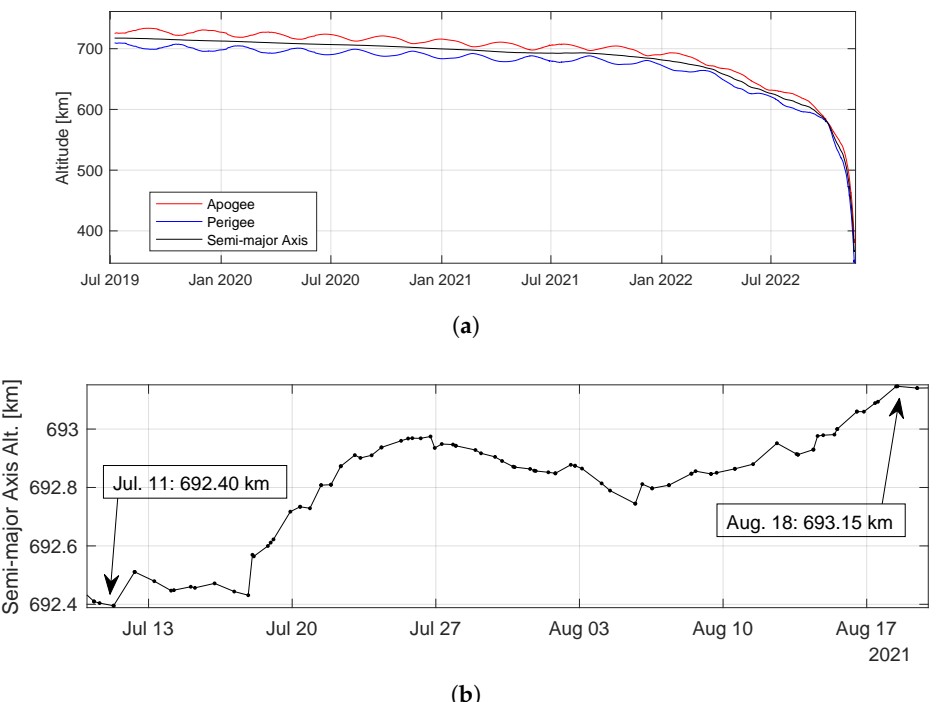

(**a**)

(**b**)

**Figure 8.** Orbit evolution throughout LS2's mission. (**a**) Overall. (**b**) Zoomed to show orbit raising during July and August 2021.

The second notable feature in Figure 8a is the oscillation in perigee and apogee. The period of this oscillation (about 100 days), matches the rate at which the line of apsides rotates with respect to the Sun vector. This was analyzed in [17]. In summary, the growth in eccentricity is greatest when the perigee is on the half of the orbit where LS2 moves away from the Sun because solar pressure will cause LS2 to speed up near perigee and slow down near apogee. This orientation changes due to the motion of the Sun in the Earth-centered frame and the $J_2$ effect of Earth's oblateness. Once perigee is on the sunward-moving half of the orbit, eccentricity will decrease.

### 4.1.1. Comparison to Orbit Propagators

Before discussing the orbit evolution in more detail, the limitations of TLE orbit knowledge must be acknowledged. TLEs represent the best fit trajectory for tracking data collected by the US Space Surveillance Network. The trajectory solutions are based on the SGP4 model, which does not include solar radiation pressure or attitude behavior [21]. The types of tracking data, number of observations, and lengths of the trajectory arcs used in the fit are also not published with the TLEs. These limitations prevent direct comparisons between LS2 attitude control performance and evolution of the orbit. Rather than trusting TLEs as an accurate reconstruction of LS2's orbit, we will instead consider them only as a source of general trends.

One way to assess the effect of solar sailing over the course of the mission is to demonstrate that LS2's actual on-orbit lifetime exceeded what would be predicted in the absence of sail control. To make this comparison, we use two orbit propagators: (i) NASA's Debris Assessment Software (DAS) [22], which is required by the Federal Communications Commission to demonstrate compliance with debris requirements for radio licensing, and (ii) General Mission Analysis Tool (GMAT) [23], a tool developed by Goddard Space Flight Center with heritage for operational orbit determination. In both propagators, the initial condition is set to the Keplerian elements derived from LS2's TLE on 23 July 2019 at 20:04:34 UTC. This epoch is less than 2 h after the sail deployment.

Both propagators assume a constant drag coefficient of 2.2 but adjust the reference area for the drag to represent the long-term average of a completely random attitude. This average is computed by assuming a uniform distribution of $\theta$, the angle between the sail normal vector and free-stream velocity:

$$A_{ref} = \frac{1}{\pi} \int_0^\pi A_{sail} |\cos\theta| d\theta = \frac{2A_{sail}}{\pi} \approx 0.6366 A_{sail}. \tag{4}$$

The area of LS2's sail is nominally 32 m$^2$. However, we also consider a reduced area of 24 m$^2$ (75%) to account for the collapsed boom observed in Figure 1c. This is a coarse assumption that is highly conservative, since it is equivalent to assuming that the length of the bent deployment boom is halved, thus halving the deployed areas of the two sail segments attached to it. In GMAT, the NRLMSIS-90 atmosphere model is used along with the consolidated F10.7 cm solar flux and Ap geomagnetic indices throughout the mission from Celestrak [24]. For solar radiation pressure, a reflectivity of 0.9 ($C_r$ = 1.8) is assumed. The sail area is scaled according to Equation (4), and the additional assumption of a spherical body is made to avoid the need for an explicit attitude model.

Table 4 reports the deorbit date and time on-orbit after sail deployment for each case. The longest case (DAS with 75% of the full sail area) deorbits more than 10 months earlier than the actual spacecraft. Both GMAT cases deorbit even sooner. From these results, we conclude that controlled solar sailing over the course of LS2's mission significantly increased its orbital lifetime.

**Table 4.** Deorbit dates for ballistic simulations of LS2's orbit. Actual deorbit date: not before 16 November 2022 (1211 days).

| Propagator | Full Sail Area | 0.75 Sail Area |
|---|---|---|
| DAS | 10 July 2021 (718 days) | 2 January 2022 (894 days) |
| GMAT | 6 November 2020 (471 days) | 21 February 2021 (579 days) |

### 4.2. Orbit Raising

Next, we consider the period from June through September 2021, during which some of the best attitude knowledge and control performance over the mission occurred. This period coincided with the only time during the mission for which consistent increases in the semi-major axis of the TLEs were observed. We consider this further evidence that LS2's On–Off control strategy was having the intended effect on the orbit.

Figure 9a shows the On–Off control performance during these months in a similar manner to Figure 6 but with the pointing accuracy averaged every 3 days. Intervals with less than 50% coverage by telemetry are omitted. The asterisk indicates the memory anomaly that disrupted solar sailing for several days. Figure 9b shows the semi-major axis and B* drag coefficient derived from TLEs. Lastly, Figure 9c provides the F10.7 cm and average Ap indices over the time frame.

The initial downward trend in the semi-major axis in Figure 9b was arrested in mid-June around the same time there was a stretch of very good On–Off control. Over the next month, rising solar flux (F10.7 cm), poor attitude control, and the memory overload anomaly appear to have resulted in additional losses in the semi-major axis. Good control returned in mid-July after recovering from the anomaly. It was around this time that the semi-major axis climbed by nearly 600 m over the course of 10 days. This was in spite of the high solar activity seen in Figure 9c. Although explaining this increase solely as the result of On–Off control is complicated (see Section 4.1.1), it is significant that the B* drag coefficient during this time sustained a negative value. Since a negative drag coefficient is non-physical, this must be due to solar radiation pressure on LS2 (which SGP4 model lacks) aliasing into the drag model. Thus, despite the limitations of SGP4, it can be concluded that the best fit for LS2's orbit during mid-late July 2021 is one that is gaining net energy with time.

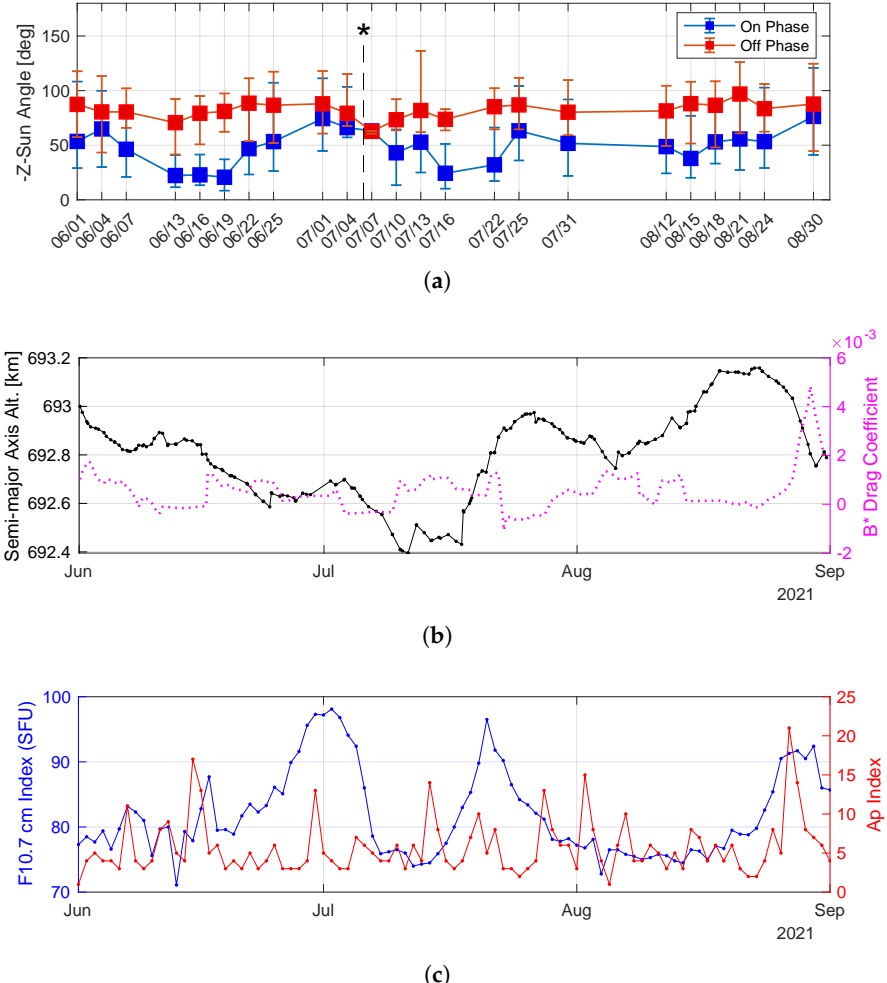

**Figure 9.** Orbit raising during summer 2021. (**a**) On–Off control performance. (**b**) Semi-major axis trend and B* drag coefficient derived from TLEs. (**c**) Space weather indices during the above period of solar sailing. (**\***) Memory overload anomaly.

LS2 briefly lost altitude before reestablishing an upward trend in the semi-major axis that was maintained until mid-August. The net gain in the TLE semi-major axis between 11 July and 22 August 2021 was 758 m. This modest gain appears to be the result of several factors:

- Spacecraft limitations: Even the lowest median pointing errors in Figure 9a are still as high as 20 deg, which is only slightly greater than the level of attitude knowledge during this time (see Figure 5). With better attitude sensors such as a star tracker and additional momentum wheels, LS2 could have achieved better control performance more consistently. A full set of reaction wheels would also have allowed LS2 to desaturate momentum continuously rather than needing to interrupt On–Off slewing for three orbits every other day to detumble.
- Reduced altitude: LightSail 2's orbit had already decayed below 700 km by this point in the mission. Had the level of control performance in Figure 9a occurred earlier, larger gains could have been achieved due to lower atmospheric density at higher altitudes.
- Solar activity: The F10.7 cm flux was not especially low during July and August 2021, reaching nearly 100 SFU, as shown in Figure 9c. Earlier periods of the mission had long stretches of quiescent space weather, which may have allowed for longer periods of sustained orbit raising, had control performance been comparable to Figure 9a.

Unfortunately, solar activity continued to increase after August 2021. The effect of space weather on the orbit is the subject of the next section.

### 4.3. Impact of Solar Activity

It is fortunate that the gyro re-calibration strategy was implemented in time to demonstrate net energy gains during the summer of 2021. Towards the end of the year, solar activity increased significantly as part of solar cycle 25. This is visible in the plot of the F10.7 cm solar flux in Figure 10a. The result was increased atmospheric drag on the spacecraft that accelerated orbit decay and prevented further periods of orbit raising despite good control performance until at least January 2022.

Figure 10b plots LS2's orbit decay rate from deployment to when the altitude reached 600 km to show how responsive it was to the F10.7 cm index. During the first two years of the mission, the F10.7 cm index remained below 100 solar flux units (SFU), and the orbit decay rate was below 100 m/day, even during periods of very poor sail control (see Figure 6). An exception is the isolated peak in solar activity in late 2020, with a correspondingly brief increase in orbit decay. Following the period of orbit raising in summer 2021, the peaks in F10.7 cm became higher and more frequent. Deeper troughs in the decay rate occur towards the right of Figure 10b and consistently line up with the peaks in Figure 10a. While this correlation was not unexpected, it serves to illustrate the large impact that atmospheric drag and space weather had on LS2's orbit performance.

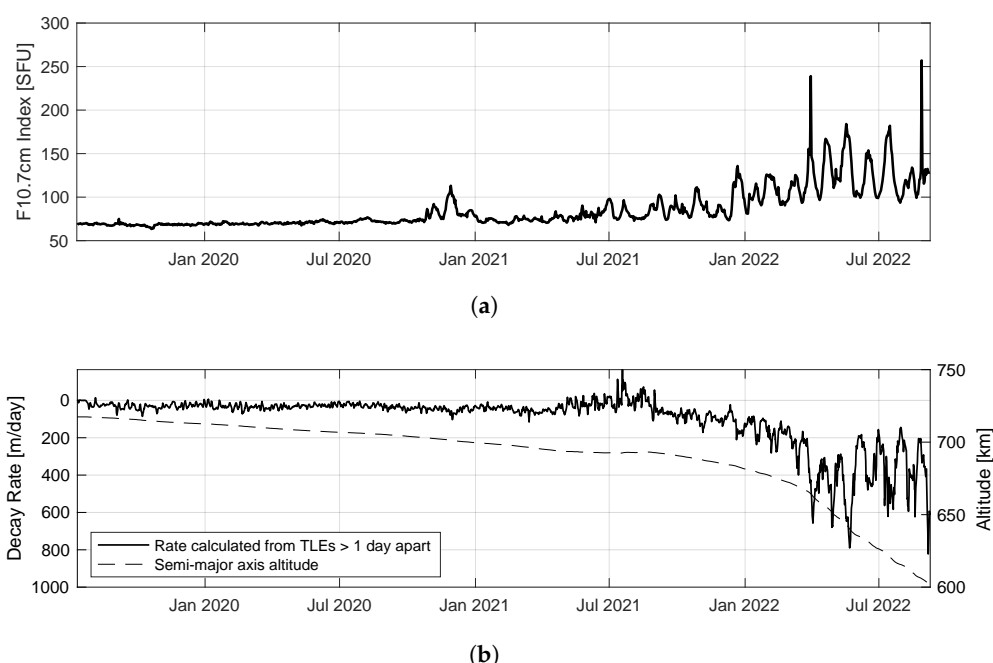

(a)

(b)

**Figure 10.** Orbit decay rate and correlation with solar activity. (**a**) Solar F10.7 cm index. LS2 launched near solar minimum, and activity increased significantly towards the end of the mission. (**b**) Orbit decay rate derived from differencing adjacent TLEs at least 1 day apart.

## 5. Deorbit

### 5.1. Data Collection

LS2's orbital altitude entered a rapid decline in early 2022. Although some periods of On–Off control comparable to the best performance of the mission (see Figure 6) were still visible at the start of the year, the effects on the orbit were overwhelmed by atmospheric drag. Difficulties keeping the PIB gyros calibrated due to challenges in downlinking sufficient 5 s telemetry data led to degrading attitude performance and, by summer 2022, On–Off slews were no longer visible in downlinked data. The team recognized that the spacecraft was no longer under control and that attitude knowledge was unreliable.

During the final months before deorbit, emphasis was placed on frequent imaging of the sail and recording the uncontrolled behavior of the satellite. In particular, we were interested in observing whether the sail would stabilize aerodynamically during the final days in orbit. The hypothesis was that the offset of the 3U CubeSat bus (and thus center of mass) from the sail might cause the sail to stabilize into a maximum drag orientation with the bus ahead of the sail. Demonstrating such behavior is relevant to drag sail research, as some proposed designs seek to leverage aerodynamic stability to accelerate orbit decay [25]. A secondary question was whether the sail might distort or collapse prematurely due to the compromised support boom.

Achieving the desired observations required a final software update. To give LS2 the best chance of stabilizing, we needed to deactivate the spacecraft's actuators so that they would not interfere with the aerodynamic torques that might provide stability. Normally this would be accomplished using LS2's "No Torques" mode (Table 3), but this mode also turned off the PIB Gyros. Since one indication of aerodynamic stability is a rotation rate that decreases with time, data from the PIB gyros were needed [25].

With the orbit decaying rapidly and two-way communication with the satellite difficult, a major software update was not feasible. Instead, our solution was to uplink zeros for the ACS control gains, transition to detumble mode to stop the momentum wheel, then return LS2 to mode 2. This successfully prevented the torque rods and momentum wheel from actuating while retaining telemetry from the PIB gyros.

Figure 11 shows the angular rates measured by the PIB gyros during the final week of the mission after the actuators had been pacified. The data were captured from telemetry beacons broadcast by LS2 every 7 s rather than coming from downlinked telemetry files. The clusters of data points therefore represent times when the spacecraft was in direct view of either the Purdue University or Cal Poly San Luis Obispo ground stations. Note that the discretization of the of the PIB rates is coarser in the beacons than in the telemetry log files.

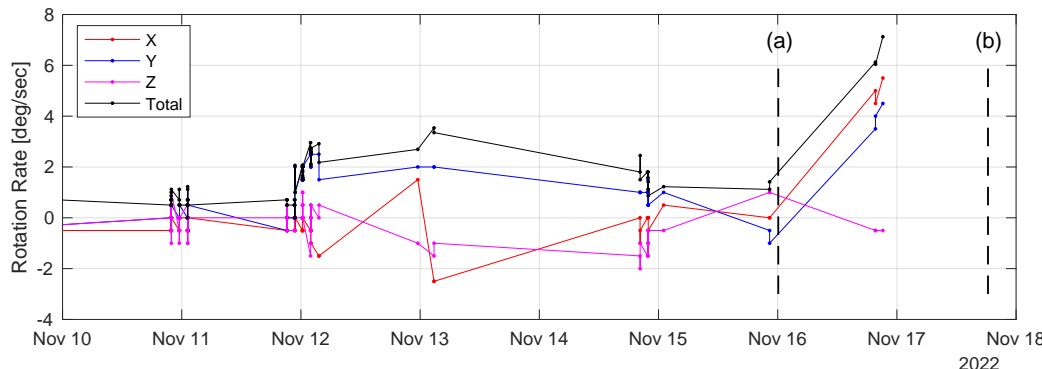

**Figure 11.** LS2 body rotation rates measured by the primary gyros during the final week in orbit. (a) Time tag of the final two-line element set. (b) Expected rise time at the Purdue ground station.

On 11 November, the rotation rates were comparable to other times during the mission. By 12 November, the total angular rate exceeded 2 deg/s and remained at this level for the next several days. During the ground station passes on 15 and 16 November, the rates had decreased. The sparsity of the data and lack of a reliable attitude estimate make it difficult to infer that LS2 had indeed stabilized; however, the decreasing rates may indicate that some aerodynamic damping was taking place. Note that, while the difficulty of performing gyro recalibrations by this time in the mission means that unknown rate biases are likely present in the data, we estimate that the biases are small relative to the trends in Figure 11. This is based on the observation that the biases measured during the first recalibration (Figure 7) were all less than 0.5 deg/s.

The final TLE epoch was at 00:06:41 UTC on November 16 and placed LS2 at an altitude of 382 km. The spacecraft was still in orbit during the tracking station passes later that day, but the observed angular rates exceeded 6 deg/s and appeared to be increasing. The last data from LS2 were received at 21:10:33 UTC from the Purdue tracking station. No

data were received during Cal Poly passes rising at 22:42 UTC November 16 or 00:19 UTC 17 November, but this was not unusual given the low elevation of these passes (<10 deg) and similar behavior on previous days. However, beacons from LS2 were not received after the expected rise time at Purdue at 18:20 UTC 17 November, or any pass thereafter.

Given the high rotation rates observed in the final beacons on 16 November and the lack of contact on 17 November, we conclude that LS2 deorbited some time between 21:10 UTC 16 November and 18:20 UTC 17 November.

### 5.2. Comparison to 6DOF Simulations

The unusually high rotation rates observed on 16 November showed that LS2 was not aerodynamically stabilized during the final set of passes. A natural question is whether these high rates indicate that the solar sail had already collapsed. To answer this question, we performed a 6-degrees-of-freedom (6DOF) Monte-Carlo simulation of LS2's orbit and attitude from the state indicated by the last TLE to deorbit.

To integrate the equations of motion, we used "VSim", a rigid-body 6DOF propagator produced by Vestigo Aerospace Inc., La Cañada Flintridge, CA, USA, for analyzing drag sail designs. The software simulates gravity gradient, solar radiation pressure, and aerodynamic torques on a panelized representation of the spacecraft. Figure 12 depicts the panel mesh and normal vectors used to represent LS2. The assumed optical properties of the sail are consistent with [26], and the simulator includes an EarthGRAM implementation of the NRLMSISE-00 atmospheric model [27]. The initial state for the orbit is derived from the final TLE, with uniform distributions of [−0.1, 0.1] deg/s for each body axis rotation rate.

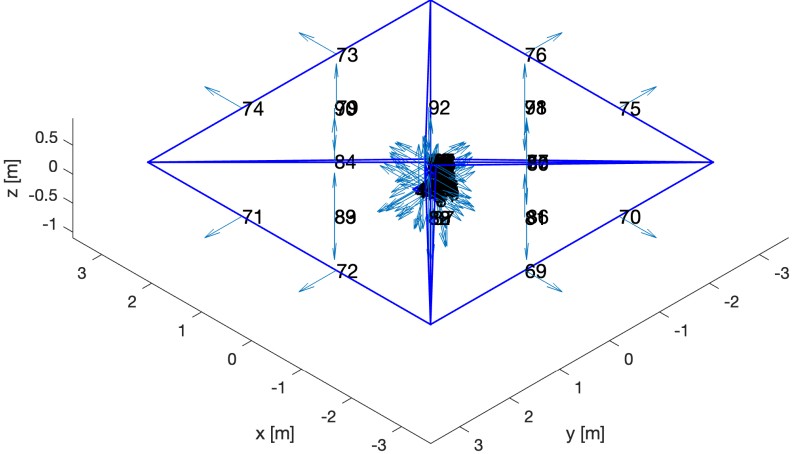

**Figure 12.** LS2 panel model for deorbit simulation.

Similar to Section 4.1.1, we consider two cases to bound the uncertainty caused by the compromised sail boom: one scenario with the full sail area, and another with 75% of the area created by shortening one of the booms by half its length (thus halving the area of the panels adjacent to that boom). We ran 250 simulations of each case. Figure 13 consolidates these runs and plots every angular rate at every time step for the full area case (Figure 13a) and the reduced area case (Figure 13b).

Figure 13a reveals that, with the full area of the sail, LS2 never achieves the high rotations rates (>5 deg/s) that were observed in the final tracking passes. In Figure 13b, the asymmetry of the sail due to the collapsed boom leads to higher rotation rates. Rates >2 deg/s are present in some of the runs more than 48 h before deorbit, while rates of 5 deg/s begin to appear 24 h before. The highest observed rate of 7.1 deg/s in the final LS2 beacon is achieved by some of the runs about 5 h prior to deorbit.

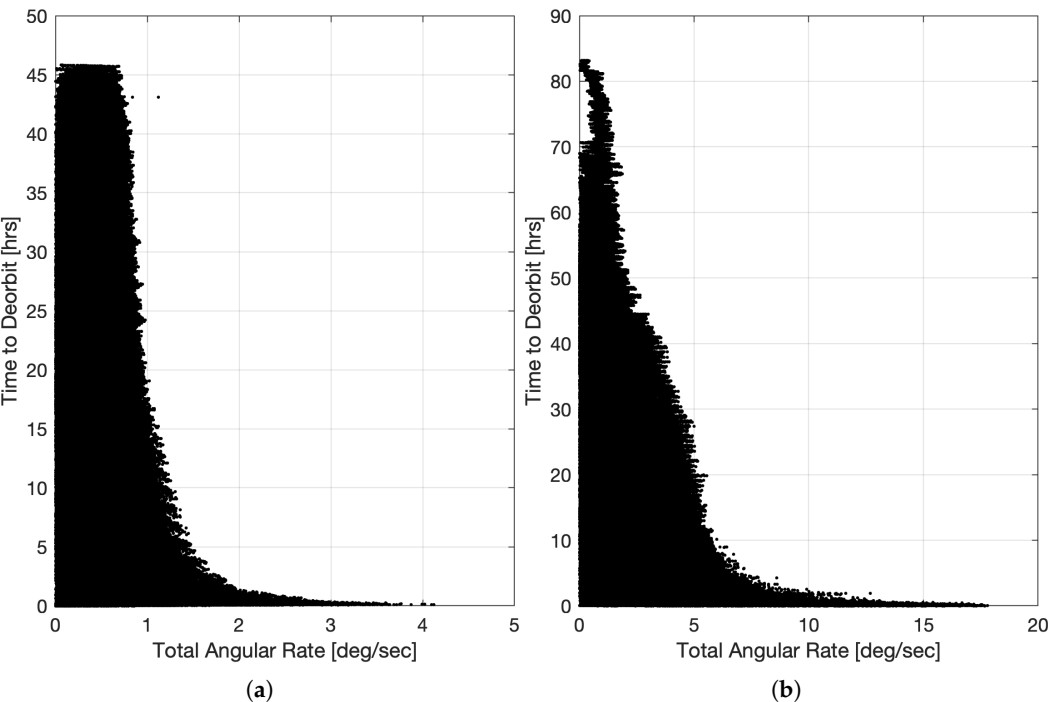

**Figure 13.** Monte Carlo simulations of LS2 attitude rates during the final deorbit. (**a**) 250 runs of the full sail area. (**b**) 250 runs with the length of one of the support booms halved.

Two conclusions can be made from this analysis. First, sail collapse is not necessary to explain the high rotation rates at the end of LS2's mission because simulations with a shortened boom length exhibit similar rates to those observed both immediately prior to deorbit and several days before. Second, the fact that rates >7.1 deg/s are achieved only a few hours before deorbit in Figure 13b suggests that LS2 deorbited within a few hours of the final tracking passes on 16 November 2022. The former conclusion is a consequential result for drag sail deorbit technology, as it suggests that even a partially failed support boom can maintain the sail until atmospheric entry is imminent.

## 6. Conclusions

The Planetary Society's LightSail 2 mission was one of only a handful of solar sails that have flown to date. It was the second controlled solar sail and the first to demonstrate control of a 3-axis stabilized sail. The combination of a 4.93 kg CubeSat bus and a 32 m$^2$ sail gave LS2 the lowest sail loading of any solar sail so far deployed in space, but also presented several challenges that had to be overcome. In this article we have analyzed how LS2 used an "On–Off" slewing strategy to affect the orbit in a controlled manner using solar radiation pressure. The limited actuators and attitude sensors and a bent deployment boom made maintaining control difficult, but performance improved throughout the flight as the team refined the momentum management strategy and performed on-obit recalibrations of the gyros.

For most of the mission, the orbit decayed due atmospheric drag, but our analysis shows that the On–Off strategy extended LS2's orbit lifetime by approximately 10 months. During the summer of 2021, LS2 also demonstrated consistent increases in its orbit semi-major axis and a negative SGP4 B* drag coefficient due to improvements in On–Off control performance. Increasing solar activity and degraded communications eventually resumed orbit decay. LS2 was allowed to tumble freely during its final week in orbit, during which we observed heightened rotation rates that were nonetheless insufficient to suggest that the weakened sail had collapsed.

LS2's orbit behavior is worthy of further analysis. A major limitation of this study has been a reliance on TLEs for defining the orbit. The limitations of the SGP4 model used to

generate TLEs meant that only general trends between LS2's orbit evolution and attitude behavior could be drawn. Future work should attempt to reconstruct the orbit with a more complete dynamic model to estimate the solar radiation pressure on the sail, including how it compares to theoretical models and how it changed over the mission. In these efforts, the TLEs could be treated as pseudo-observables to offset the lack of raw observations for orbit determination. It may also be wise to constrain uncertainties in atmospheric density and the resulting drag on LS2 by simultaneously fitting the orbits of other spacecraft with the model. Ideally, these other spacecraft should be those for which raw observations are available.

Finally, our analysis has focused mainly on the evolution of LS2's semi-major axis. The effect of LS2's solar sailing on other orbit elements is currently unexplored and could be important for extracting empirical sail accelerations. Obtaining these results will help validate the solar radiation pressure models that are necessary to design solar sail trajectories. Having demonstrated controlled solar sailing, the next milestone for this technology is to fly a trajectory, such as an artificial Lagrange point, which is uniquely suited to solar sails. This will prove the practical benefits of solar sailing and help realize their ultimate potential.

**Author Contributions:** Conceptualization, B.B.; methodology, All; software, J.M.B., B.P. and J.R.M.; validation, All; formal analysis, J.R.M., D.A.S. and B.B.; investigation, All; resources, J.M.B. and D.A.S.; data curation, J.M.B., J.R.M. and B.B.; writing—original draft preparation, J.R.M.; writing—review and editing, J.R.M. and D.A.S.; visualization, J.R.M.; supervision, B.B. and D.A.S.; project administration, B.B. and D.A.S. All authors have read and agreed to the published version of the manuscript.

**Funding:** This research received no external funding.

**Data Availability Statement:** Telemetry beacon data for the entire LS2 mission can be downloaded from the LightSail 2 Mission Control page at https://secure.planetary.org/site/SPageNavigator/mission_control.html (accessed on 20 June 2023).

**Acknowledgments:** The authors would like to thank the donors and members of The Planetary Society as well as the Kickstarter campaign contributors who supported the LightSail program. We are grateful for the crucial assistance of the University Nanosat Program, Air Force Research Laboratory, the Prox-1 team, the U.S. Department of Defense Space Test Program, and the U.S. Air Force 18th Space Control Squadron. We also thank NASA's Near Earth Asteroid Scout team for informative discussions and a long-term collaboration through a NASA Space Act Agreement. Finally, we are thankful for the assistance of Anthony Cofer of Purdue University for his help in maintaining the tracking equipment.

**Conflicts of Interest:** The authors declare no conflict of interest. The funders had no role in the design of the study; in the collection, analyses, or interpretation of data; in the writing of the manuscript; or in the decision to publish the results.

## Abbreviations

The following abbreviations are used in this manuscript:

| | |
|---|---|
| 6DOF | 6 Degrees of Freedom |
| ADCS | Attitude Determination and Control System |
| DAS | Debris Assessment Software |
| GMAT | General Mission Analysis Tool |
| IGRF | International Geomagnetic Reference Field |
| LS2 | LightSail 2 |
| SFU | Solar Flux Units |
| SGP4 | Simplified Generalized Perturbations |
| TRAC | Triangular Rollable And Collapsible |
| TRL | Technology Readiness Level |

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
