# Peer review of "LightSail 2 Solar Sail Control and Orbit Evolution"

_aerospace, doi:10.3390/aerospace10070579_

Round 1
Reviewer 1 Report
LightSail 2 mission is a significant technical breakthrough mission for solar sailing. This paper reviews the solar sail control and the effect on orbit throughout the LightSail 2 mission operations, which provide valuable insights for future solar sailing missions. The authors provide details of the solar sail control system design, operations scheme, and telemetry analysis results to discuss the solar sailing performance of LightSail 2. Considering the significance of the mission, the in-depth analysis of various practical aspects for future solar sail missions, the paper meets the publication standards. Yet, there are few questions on comments throughout the paper that may improve the quality of the paper.
Page 1, line 29:
Please be specific with “Since the thrust of a solar sail scales with the inverse square law, they are particularly advantageous for transfers and stationkeeping within the inner solar system.” An extra line explaining how the solar sail helps matching the required rate of change of the ascending node angle in Mercury missions will help understand the potential application of solar sailing in inner solar system missions.
Page 1, line 34:
“solar pressure” should be solar radiation pressure, as it may be misleading to pressure rather than force.
Page 3, Figure 1c:
Is it possible to estimate the effective solar sail area using the solar sail images?
Were the sails bent in the same shape during On/Off operations?
Furthermore, was the 75% solar sail area (24 m^2) throughout the paper derived from Figure 1c or was it a coarse assumption?
Page 5, Table 3:
What kind of control algorithms and attitude filters were used in the different modes? The full details do not have to be described, but it would be helpful for the readers to understand the system performance if the implemented algorithms are stated.
Page 6, line 167:
Small typo in “usesdwhen.”
Page 7, Figure 3:
How was the attitude estimated in the eclipse, as an On to Off transition is performed inside the eclipse according to Figure 2? Could this be explained somewhere in the manuscript?
It is a minor comment, but it would be easier to recognize the subplots with subtitles.
Page 13, Figure 8b:
Using such telemetry data, is it possible to estimate the SRP range? If so, SRP can be compared against theoretical values. It would be interesting if changes in reflectivity over time, or membrane shape deformation depending on attitude could be estimated for further insights in solar sailing.
Page 13, line 365:
“Note that this effect is not related to the On-Off control strategy.”
The variations of orbital elements will depend on how the orbital elements and the On-Off control strategy coincides, which implies that the orbit perturbation is effected by the On-Off control strategy. The sentence above is confusing regarding this point.
Page 18, Figure 12:
As VSim is a rigid-body propagator, it does not solve sail membrane dynamics. Will the Figure 13a results show a better stabilized performance if the sail membrane bends to form a conical wedge-like shape? Is it possible to estimate the bending angle of the booms were it not to be an anomaly on the sail at lower altitudes?
Page 19, Conclusion:
Based on LightSail 2 results, what would be the next milestone regarding the advance of solar sail technology?
Author Response
Thank you very much for your insights and helpful suggestions. Please find our responses attached.

Reviewer 2 Report
See attached.

see attached.
Author Response
Thank you very much for your insights and comments. Please find our responses attached.

Round 2
Reviewer 2 Report
accept